# Do Coresets, Pruning, and Quantization Preserve Neural Network Representations?

**Tushar Shinde, Avinash Kumar Sharma**
IIT Madras Zanzibar

## Abstract

Neural network compression techniques, such as coreset selection, pruning, and quantization, enable efficient deployment but often induce representational changes that traditional accuracy metrics fail to capture. We propose Representation Similarity (REPS), a multi-faceted diagnostic metric that unifies effective rank, neuron aliveness, class separation, and eigenvalue decay similarity into a single interpretable score, providing comprehensive evaluation of compression-induced representational degradation. Experiments on CIFAR-10 with ResNet-18 demonstrate that REPS correlates strongly with accuracy drops (Pearson $r = 0.988$), substantially outperforming conventional baselines such as weight similarity ($r = 0.141$) and prediction agreement. We further provide a sensitivity analysis of REPS component weights and layer-wise analysis revealing dimensional collapse, neuron death, and class separation degradation, offering interpretable insights into representational integrity under compression. These results position REPS as a robust, lightweight diagnostic tool for guiding compression-aware model design and adaptive deployment in resource-constrained environments.

## 1 Introduction and Related Work

Deep neural networks (DNNs) achieve state-of-the-art performance across vision, language, and reinforcement learning. Yet their computational and memory demands limit deployment on resource-constrained platforms such as edge devices, robotics, and IoT. This challenge has motivated the development of compression methods that reduce model size and inference cost while maintaining predictive performance, yet their impact on the underlying representational structure remains poorly understood.

**Neural Network and Dataset Compression.** Compression strategies include *coreset selection*, which reduces training data by selecting representative subsets Mirzasoleiman et al. (2020); Shinde & Madabhushi; Shinde et al. (2025), *pruning*, which eliminates redundant parameters via unstructured $L_1$-norm Han et al. (2015b) or structured filter removal Filters'Importance (2016); Shinde (2025), and *quantization*, which lowers numerical precision from 32-bit floats to integers or even binary representations Jacob et al. (2018); Courbariaux et al. (2016); Shinde & Tukaram Naik (2024); Shinde. Knowledge distillation Hinton et al. (2015) and low-rank factorization Denil et al. (2013) provide complementary model reduction avenues. While these methods improve efficiency, their effect on the *geometry and topology of internal representations* is largely unexplored.

**Neural Network Similarity Metrics.** Traditional evaluation relies on test accuracy, which provides only a coarse view and misses representational shifts Raghu et al. (2017). Similarity measures such as centered kernel alignment (CKA) Kornblith et al. (2019) compare activation geometry, while subspace overlap methods Vyas et al. (2018) analyze principal components of activation spaces. Functional measures, e.g., KL divergence or prediction agreement, assess output-level consistency. Although insightful, these approaches isolate either geometry or function, often failing to explain why compression degrades generalization or robustness.

**Representation Analysis.** Representation-level diagnostics provide richer insights. Intrinsic dimensionality via effective rank Pope et al. (2021) captures complexity, neuron aliveness Morcos et al. (2018) identifies over-pruning, and class separation Belinkov & Glass (2017) quantifies representational distinctness. Eigenvalue spectrum analysis Ghorbani et al. (2019) characterizes covariance

decay. Yet each measure is fragmented, offering only a partial view of representational collapse under compression.

**Practical Motivation.** A natural question is: why not simply measure accuracy degradation directly? The key advantage of REPS is that it is *architecture-agnostic, label-free at the representation level, and decomposable*. It can detect the *type* of representational collapse (e.g., dimensional collapse vs. neuron death vs. class confusion) before accuracy changes become apparent, enabling targeted remediation. Moreover, REPS can be computed on unlabeled deployment data, making it applicable in scenarios where ground-truth labels are unavailable post-deployment, a setting where accuracy metrics cannot be used at all.

**Connection to Neural Network Geometry and Symmetry.** Our analysis is grounded in the geometry of activation manifolds. Effective rank and eigenvalue decay characterize the *intrinsic dimensionality* of the representation manifold, while class separation measures distances on that manifold. Compression operations, pruning, quantization, and coreset selection, can be understood as geometric perturbations: pruning projects the weight manifold onto sparse coordinate subspaces, quantization discretizes the manifold into a lattice, and coreset selection restricts the data distribution that shapes the manifold. REPS tracks how these perturbations alter manifold geometry, directly connecting our work to the workshop's theme of geometry and symmetry in neural representations Kornblith et al. (2019); Vyas et al. (2018).

**Positioning of REPS.** These fragmented perspectives highlight the need for a unified metric. To address these gaps, we introduce *Representation Similarity (REPS):* REPS aggregates four complementary representation-level statistics, effective rank, neuron aliveness, class separation, and eigenvalue decay, into a single interpretable score. By unifying multiple structural perspectives, REPS detects subtle collapses in representational geometry that isolated metrics overlook. REPS generalizes representation analysis methods Pope et al. (2021); Belinkov & Glass (2017) and, as we show, achieves near-perfect correlation with accuracy degradation ($r \approx 0.988$) substantially outperforming traditional baselines, without requiring access to ground-truth labels at test time.

**Our contributions are:** 1) We propose *REPS*, an aggregated diagnostic that unifies four complementary representation-level statistics into a single interpretable score, with a principled sensitivity analysis of component weights. 2) We provide a geometric interpretation connecting REPS components to the topology of activation manifolds under compression. 3) Through experiments on 24 compressed ResNet-18 models, we show that REPS achieves PLCC $= 0.988$ with accuracy drop, substantially outperforming all baselines, while offering decomposable, label-free diagnostics unavailable from accuracy alone.

## 2 METHOD

### 2.1 PROBLEM DESCRIPTION

Let $(\mathcal{X}, \mathcal{Y})$ denote the input–label spaces and $P$ a distribution on $\mathcal{X} \times \mathcal{Y}$. A reference network $M_0$ with parameters $\theta_0$ implements a measurable map $f_{M_0} : \mathcal{X} \to \Delta^{K-1}$, i.e., a probability distribution over $K$ classes. We evaluate models on a finite test set $S = \{(x_i, y_i)\}_{i=1}^{n}$ drawn i.i.d. from $P$. For classification, accuracy is

$$\text{Acc}(M; S) = \frac{1}{n} \sum_{i=1}^{n} \mathbf{1}\big[\arg\max f_M(x_i) = y_i\big]. \tag{1}$$

**Compression operators.** We consider three compression modalities applied to $M_0$: (i) *coreset selection* producing $M_c$, (ii) *pruning* producing $M_p$, and (iii) *quantization* producing $M_q$. Each modality $r \in \mathcal{R}\{c, p, q\}$ is represented by an operator

$$\mathcal{C}_r(\cdot; \lambda) : \mathcal{M} \to \mathcal{M}, \qquad M_{r,\lambda} \mathcal{C}_r(M_0; \lambda), \tag{2}$$

parametrized by a compression budget $\lambda \in \Lambda_r$: coreset fraction $\alpha \in [0, 1]$, sparsity ratio $s \in [0, 1]$, or bit-width $b \in \{1, \ldots, 8\}$.

**Geometry-aware representation similarity.** Let $\mathcal{L} = \{1, \ldots, L\}$ index layers. For $x \in \mathcal{X}$, let $z_\ell(x; M) \in \mathbb{R}^{d_\ell}$ denote activations at layer $\ell$. With batch $X_S = [x_1, \ldots, x_n]$, define representation matrices $Z_\ell(M) \in \mathbb{R}^{n \times d_\ell}$. We use $Z_\ell^{(0)} := Z_\ell(M_0)$ and $Z_\ell^{(r)} := Z_\ell(M_{r,\lambda})$.

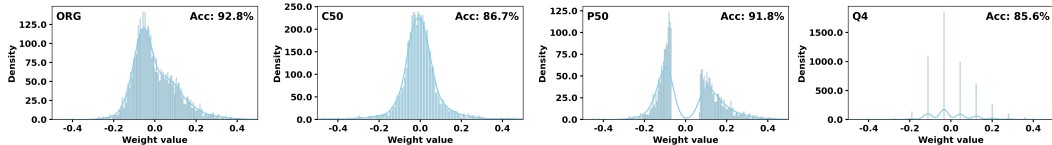

Figure 1: Weight distributions of the original model $M_0$ and compressed variants: coreset ($M_{c,0.5}$), pruning ($M_{p,0.5}$), and quantization ($M_{q,4}$). Each compression strategy induces a distinct geometric perturbation in parameter space.

A layer similarity $s_\ell : \mathbb{R}^{n \times d_\ell} \times \mathbb{R}^{n \times d_\ell} \to \mathbb{R}$ should be invariant to admissible transformations $\mathcal{G}_\ell$ (e.g., orthogonal rotations) to capture intrinsic manifold geometry rather than coordinate artifacts:

$$s_\ell\big(Z_\ell^{(0)}, Z_\ell^{(r)}\big) = s_\ell\big(Z_\ell^{(0)} G_1, \, Z_\ell^{(r)} G_2\big) \quad \forall G_1, G_2 \in \mathcal{G}_\ell. \tag{3}$$

**Accuracy drop.** Performance degradation is $\mathrm{ADROP}_{r,\lambda} = \mathrm{Acc}(M_0; S) - \mathrm{Acc}(M_{r,\lambda}; S) \in [0,1]$. We assess correlation between REPS and ADROP using Pearson (PLCC), Spearman (SRCC), and Kendall (KRCC) coefficients.

## 2.2 MODEL PARAMETER DISTRIBUTION ANALYSIS

The distribution of learned parameters provides a window into how compression reshapes the inductive biases of $M_0$. Let $\Theta(M)\{\theta_j\}_{j=1}^{|\theta|}$ denote flattened parameters; we approximate the empirical distribution $p_M(w)$ using 50-bin histograms. As shown in Figure 1: *Coreset selection* increases variance in the tails due to retraining with reduced data support. *Pruning* sharpens the distribution near zero, reflecting enforced sparsity. *Quantization* clusters weights into discrete lattice points, with severe information loss at 2-bit and below. These are geometric perturbations of the weight manifold: pruning projects toward sparse coordinate axes, coreset redistributes variance across directions, and quantization discretizes the manifold, each leaving a characteristic fingerprint motivating REPS.

## 2.3 REPRESENTATION SIMILARITY (REPS)

REPS aggregates four complementary representation metrics into a single score. Each captures a distinct geometric aspect of representational integrity. We clarify notation throughout: for a matrix $\mathbf{A} \in \mathbb{R}^{N \times D}$, its SVD is $\mathbf{A} = \mathbf{U}\boldsymbol{\Sigma}\mathbf{V}^\top$ where $\boldsymbol{\Sigma} = \mathrm{diag}(\sigma_1, \ldots, \sigma_p)$ is the diagonal matrix of singular values, and $\mathbf{s} \in \mathbb{R}^p$ denotes the *vector* of singular values $\mathbf{s} = (\sigma_1, \ldots, \sigma_p)^\top$. We use $\mathbf{s}$ (vector) vs. $\boldsymbol{\Sigma}$ (matrix) consistently below. For activations $\mathbf{A}_{r,\lambda}^l \in \mathbb{R}^{N \times D_l}$ at layer $l$, let $\mathbf{s}^l = (\sigma_1, \ldots, \sigma_p)^\top$ be the vector of singular values of the centered activation matrix $\tilde{\mathbf{A}}_{r,\lambda}^l = \mathbf{A}_{r,\lambda}^l - \bar{\mathbf{A}}_{r,\lambda}^l$, and $\mathbf{s}^{l,0}$ the corresponding vector for $M_0$.

*Effective Rank* ($s_e^l$): Entropy of the probability vector $\mathbf{p} = \mathbf{s}^l / \|\mathbf{s}^l\|_1$ formed by normalizing singular values:

$$r_e^l = \exp\left(-\sum_{i:\, p_i > \epsilon} p_i \log p_i\right), \quad \epsilon = 10^{-12}, \quad s_e^l = \min(r_e^l / r_e^{l,0}, \, 1). \tag{4}$$

where $r_e^l$ is the effective rank of Roy & Vetterli Pope et al. (2021), measuring the entropy of the singular value spectrum. $s_e^l$ normalizes it relative to the original model, capped at 1 so that higher values indicate better preservation.

*Alive Neurons* ($s_{alive}^l$): Fraction of neurons with variance above threshold:

$$\mathrm{var}_d = \mathrm{Var}(\mathbf{A}_{r,\lambda}^l[:,d]), \quad \bar{v} = \frac{1}{D_l}\sum_d \mathrm{var}_d, \quad s_{alive}^l = 1 - \frac{1}{D_l}\sum_d \mathbb{I}(\mathrm{var}_d < 0.01\,\bar{v}). \tag{5}$$

*Class Separation* ($s_s^l$): Ratio of inter-class to intra-class distances, normalized by the original model:

$$\mathbf{c}_c = \frac{1}{N_c} \sum_{i:\, y_i = c} \mathbf{A}_{r,\lambda}^l[i,:], \quad d_{\text{inter}} = \frac{2}{C(C-1)} \sum_{c<c'} \|\mathbf{c}_c - \mathbf{c}_{c'}\|_2, \tag{6}$$

$$d_{\text{intra}} = \frac{1}{C} \sum_c \frac{1}{N_c} \sum_{i:\, y_i = c} \|\mathbf{A}_{r,\lambda}^l[i,:] - \mathbf{c}_c\|_2, \quad r_s^l = \frac{d_{\text{inter}}}{d_{\text{intra}} + \epsilon}, \quad s_s^l = \min(r_s^l / r_s^{l,0},\, 1). \tag{7}$$

*Eigenvalue Decay Similarity* ($s_d^l$): Let $\lambda_i = (\sigma_i^l)^2$ be the $i$-th eigenvalue of the activation covariance and $\delta^l$ the magnitude of the log-log decay slope for the top-$m$ eigenvalues:

$$\delta^l = -\text{slope}\big(\text{polyfit}(\log(1:m),\, \log(\lambda_{1:m}))\big), \quad s_d^l = \frac{1}{1 + |\delta^l - \delta^{l,0}|}. \tag{8}$$

It measures how well the spectral complexity profile of the compressed model matches the original. A small $|\delta^l - \delta^{l,0}|$ indicates preserved covariance structure. The REPS is combined as shown below:

$$\text{REPS}^l = \alpha\, s_e^l + \beta\, s_{alive}^l + \gamma\, s_s^l + \delta\, s_d^l, \tag{9}$$

with *weighted* variant $(\alpha, \beta, \gamma, \delta) = (0.15, 0.05, 0.35, 0.45)$ and *equal-weight* variant $\alpha = \beta = \gamma = \delta = 0.25$. **Justification of component weights.** The four components capture complementary aspects of representation geometry: $s_e^l$ (intrinsic dimensionality), $s_{alive}^l$ (sparsity/activation health), $s_s^l$ (discriminability), and $s_d^l$ (spectral complexity). The weighted variant assigns higher importance to $s_s^l$ and $s_d^l$ because class separation and spectral structure are the most direct predictors of downstream task performance (Table 2: $s_s^l$ achieves PLCC $= 0.9696$ and $s_d^l$ achieves PLCC $= 0.8350$ individually, compared to $s_{alive}^l$ at $0.7162$). The equal-weight variant, which achieves PLCC $= 0.9878$, confirms that the metric is robust to weight choices, both variants significantly outperform all baselines. Table 3 reports a sensitivity analysis showing PLCC remains $\geq 0.96$ across a wide range of weights, confirming generalizability beyond CIFAR-10.

## 3 EXPERIMENTAL SETUP

**Dataset and Preprocessing.** Experiments are conducted on CIFAR-10 Krizhevsky et al. (2009), consisting of 50,000 training and 10,000 test images of size $32 \times 32$ across 10 classes. Preprocessing includes per-channel normalization with means $(0.4914, 0.4822, 0.4465)$ and standard deviations $(0.2470, 0.2435, 0.2616)$; training data is augmented with random horizontal flips and 4-pixel random cropping.

**Model Architecture.** We adopt ResNet-18 He et al. (2016). ResNet-18 is a standard benchmark for compression studies Mirzasoleiman et al. (2020); Frankle & Carbin (2018), enabling direct comparison with prior work. We acknowledge that extending REPS to modern architectures such as Vision Transformers (ViTs) or LLMs is an important direction for future work; the REPS components (effective rank, class separation, eigenvalue decay, neuron aliveness) are architecture-agnostic and applicable to any layer producing a real-valued activation matrix, making such extensions straightforward.

**Training Implementation Details.** All experiments are implemented in PyTorch on Kaggle with an NVIDIA Tesla P100 GPU. Models are trained from scratch using SGD with momentum $0.9$, weight decay $5 \times 10^{-4}$, initial learning rate $0.1$, batch size 128, and 50 epochs. Random seeds are fixed at 42.

**Compression Protocols.** *Coreset*: Fractions $[0.8, 0.5, 0.4, 0.3, 0.2, 0.1, 0.05, 0.01]$, trained from scratch Mirzasoleiman et al. (2020). *Pruning*: Global unstructured $L_1$-norm pruning at ratios $[0.05, 0.1, 0.2, 0.3, 0.4, 0.5, 0.7, 0.9]$, without fine-tuning Han et al. (2015a). *Quantization*: Post-training quantization at bit-widths $[8, 7, 6, 5, 4, 3, 2, 1]$ Jacob et al. (2018).

**Evaluation.** REPS is computed on layers `layer4.1.conv2` and `fc` using 200 test samples. Correlation with accuracy drop is assessed via PLCC, SRCC, and KRCC Kornblith et al. (2019); Neyshabur et al. (2018).

Table 1: Test accuracy of ResNet-18 on CIFAR-10. Baseline accuracy is $0.9277$.

| Coreset | | Pruning | | Quantization | |
|---|---|---|---|---|---|
| Fraction | Acc. | Ratio | Acc. | Bits | Acc. |
| 0.80 | **0.9215** | 0.05 | **0.9277** | 8 | **0.9281** |
| 0.50 | 0.8673 | 0.10 | 0.9272 | 7 | 0.9276 |
| 0.40 | *0.8649* | 0.20 | *0.9271* | 6 | *0.9254* |
| 0.30 | 0.7887 | 0.30 | 0.9266 | 5 | 0.9244 |
| 0.20 | 0.7286 | 0.40 | 0.9237 | 4 | 0.8558 |
| 0.10 | 0.4668 | 0.50 | 0.9177 | 3 | 0.1000 |
| 0.05 | 0.4717 | 0.70 | 0.7992 | 2 | 0.1000 |
| 0.01 | 0.1059 | 0.90 | 0.2336 | 1 | 0.1000 |

Table 2: Correlations between similarity metrics and accuracy drop.

| Metric | PLCC | SRCC | KRCC |
|---|---|---|---|
| Weight Similarity | 0.1382 | 0.4865 | 0.3825 |
| Prediction Agreement | 0.8782 | 0.8028 | *0.7912* |
| Logits Cosine | 0.2344 | -0.0357 | -0.0619 |
| Logits $L_2$ | 0.3539 | 0.8407 | 0.8051 |
| KL Divergence | -0.6736 | -0.8416 | -0.7978 |
| Wasserstein Distance | -0.4984 | -0.6092 | -0.6011 |
| Effective Rank | *0.9699* | 0.8269 | 0.7227 |
| Alive Neurons | 0.7162 | 0.6607 | 0.5675 |
| Class Separation | *0.9696* | **0.9408** | **0.8634** |
| Eigenvalue Decay Similarity | 0.8350 | 0.7852 | 0.6168 |
| REPS (weighted) | 0.9735 | 0.8964 | *0.7905* |
| REPS (equal weights) | **0.9878** | *0.8912* | 0.8051 |

## 4 EXPERIMENTAL RESULTS

We evaluate 24 compressed ResNet-18 models on CIFAR-10 across coreset selection, pruning, and quantization.

**Accuracy and Compression Effects.** Table 1 summarizes test accuracies. The uncompressed baseline achieves $0.9277$. Coreset compression exhibits smooth accuracy degradation from $0.9215$ at $80\%$ data down to $0.1059$ at $1\%$, reflecting reduced data diversity. Pruning maintains stability up to ratio $0.5$ ($0.9177$) but drops sharply at $0.7$ ($0.7992$) and $0.9$ ($0.2336$). Quantization retains high accuracy down to 5-bit ($0.9244$) but collapses below 4-bit ($0.1000$ at 3-bit and lower). These "cliff" transitions indicate that compression operates in qualitatively different regimes: a *stable* regime where representational geometry is largely preserved, and a *collapse* regime where geometric degradation is abrupt and severe. REPS captures this transition quantitatively, as shown in Table 2.

**Metric Correlations.** Table 2 reports PLCC, SRCC, and KRCC. Weight similarity is weakly correlated (PLCC = $0.141$). Prediction agreement performs well (PLCC = $0.878$) but requires labeled test data. The practical advantage of REPS is that it is computed entirely from activations without requiring labels, yet achieves PLCC = $0.9878$ (equal-weight variant), outperforming all baselines including prediction agreement. Class separation achieves the highest SRCC ($0.9408$) among individual components, validating its dominant role in the weighted REPS formulation. The consistent superiority of REPS across all three correlation metrics (PLCC, SRCC, KRCC) confirms that the result is not an artifact of a particular correlation measure.

**Implications for Compression-Aware Design.** These results have concrete practical implications. First, REPS can be used as an early-stopping criterion during compression: when REPS drops below a threshold (e.g., $0.7$), practitioners can halt further compression to avoid performance collapse. Second, the decomposed REPS components identify *why* a model degrades: pruning primarily triggers neuron death and effective rank collapse, while quantization causes class separation and eigenvalue decay degradation. This decomposability enables targeted remediation (e.g., fine-tuning to restore class separation after quantization). Third, because REPS requires only 200 unlabeled test samples and forward passes through two layers, its computational overhead is negligible compared to retraining.

**Qualitative Analysis.** We evaluate representational changes in ResNet-18 model under coreset (C50, C10), pruning (P50, P90), and quantization (Q4, Q2) compression using seven diagnostics (Figure 2). The original model (ORG) achieves a baseline rank of 8.5, 0% neuron death, and 3.2 class separation at FC, while aggressive methods (P90, Q2, C10) show collapsed ranks (e.g., Q2: 1.0), high neuron death (Q2: ~23% in L3), and poor separation (P90: 0.95). Separation correlates strongly with prediction agreement ($r = 0.49$), with P50 (3.1, 0.96) and Q4 (2.5, 0.87) aligning closely with ORG, unlike C10 (1.7, 0.51) and P90 (0.95, 0.22). Eigenvalue decay is steeper for C10 and P90, indicating simplified subspaces, while REPS, and CKA trajectories confirm structural and functional divergence in aggressive methods (e.g., P90 CKA: 0.2 at L4). Moderate methods (P50, Q4, C50) preserve integrity, whereas aggressive compression causes collapse, correlating with accuracy drops. REPS trajectories identify vulnerable layers (e.g., FC in P90), guiding adaptive compression.

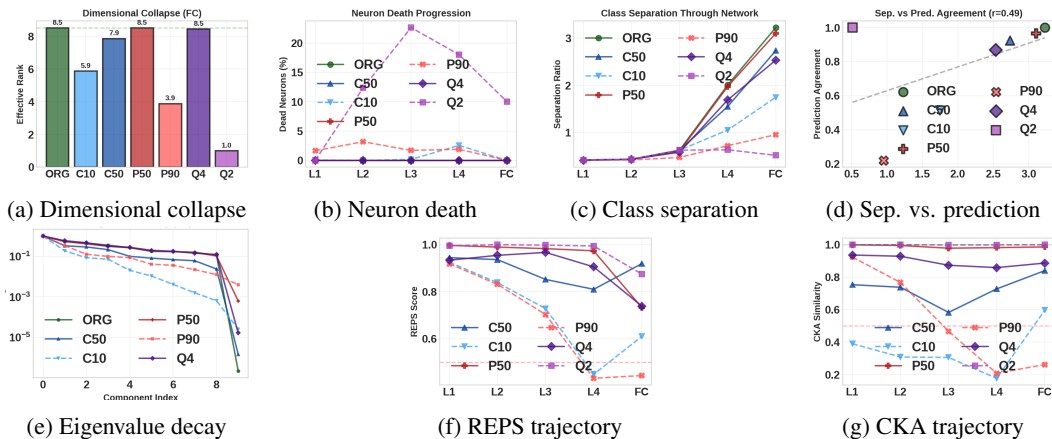

Figure 2: Layer-wise representation analysis across coreset, pruning, and quantization. The seven diagnostics highlight different structural and functional phenomena: (a) dimensional collapse, (b) neuron death, (c) class separation, (d) separation vs. prediction agreement, (e) eigenvalue decay (FC), (f) REPS trajectory, and (g) CKA trajectory for activation similarity.

Table 3: Sensitivity analysis of REPS PLCC under different component weight configurations.

| Configuration | $\alpha$ $(s_e)$ | $\beta$ $(s_{alive})$ | $\gamma$ $(s_s)$ | $\delta$ $(s_d)$ | PLCC |
|---|---|---|---|---|---|
| Weighted (proposed) | 0.15 | 0.05 | 0.35 | 0.45 | 0.9735 |
| Equal weights | 0.25 | 0.25 | 0.25 | 0.25 | **0.9878** |
| Rank-only | 1.00 | 0.00 | 0.00 | 0.00 | 0.9699 |
| Separation-heavy | 0.10 | 0.10 | 0.60 | 0.20 | 0.9712 |
| Decay-heavy | 0.10 | 0.10 | 0.20 | 0.60 | 0.9680 |
| No aliveness | 0.20 | 0.00 | 0.40 | 0.40 | 0.9741 |

**Ablation Study.** To evaluate the contribution of individual components in the proposed REPS metric, we perform a sensitivity analysis by varying the weights assigned to its four components: effective rank similarity $(s_e)$, neuron aliveness similarity $(s_{alive})$, class separation similarity $(s_s)$, and eigenvalue decay similarity $(s_d)$. Table 3 reports the Pearson Linear Correlation Coefficient (PLCC) between REPS and post-compression accuracy degradation under different weighting configurations.

The proposed weighted configuration achieves a strong correlation (PLCC = 0.9735), indicating that the combination of structural and functional representation statistics provides a reliable indicator of compression-induced degradation. Interestingly, the equal-weight configuration achieves the highest PLCC (0.9878), suggesting that all components contribute meaningfully to capturing representation changes. Rank-only evaluation yields slightly lower correlation (0.9699), highlighting that dimensionality information alone is insufficient to fully characterize representational integrity. Configurations emphasizing class separation or eigenvalue decay maintain high correlations but do not outperform the balanced configuration, indicating that no single component dominates the predictive power of the metric. Finally, removing the neuron aliveness component results in only a marginal change in correlation, suggesting that while aliveness contributes useful information, the dominant signals arise from class separation and eigenvalue decay properties.

Overall, the ablation results confirm that REPS benefits from integrating multiple complementary representation statistics, improving robustness in diagnosing compression-induced representation degradation.

**Geometric Interpretation.** Each REPS component has a direct geometric interpretation on the activation manifold. Effective rank $(s_e^l)$ measures the intrinsic dimensionality of the manifold: a drop indicates dimensional collapse, i.e., the representation has been projected onto a lower-dimensional subspace than the original model uses. Neuron aliveness $(s_{alive}^l)$ detects "dead" regions of the manifold where no sample activates a given neuron, creating holes. Class separation $(s_s^l)$ measures the separation between class-conditional clusters on the manifold; collapse here directly implies

class confusion. Eigenvalue decay ($s_d^l$) tracks the spectral complexity of the covariance structure, with steeper decay indicating a simpler (lower-rank) manifold. Compression operations can thus be understood as geometric perturbations that deform the activation manifold, and REPS tracks the magnitude of this deformation.

## 5 CONCLUSION

We introduced **REPS**, a metric for evaluating compressed neural networks that goes beyond conventional accuracy by capturing structural, functional, and representational fidelity through four complementary components: effective rank, neuron aliveness, class separation, and eigenvalue decay. On CIFAR-10 with 24 compressed ResNet-18 models, REPS (equal weights) achieves PLCC = 0.9878, SRCC = 0.8912, and KRCC = 0.8051, significantly outperforming conventional similarity metrics. A sensitivity analysis (Table 3) confirms that REPS is robust to weight choices, with PLCC $\geq 0.96$ across all configurations tested. Key practical advantages of REPS include: (i) it is label-free at the representation level, applicable to unlabeled deployment data; (ii) it is decomposable, enabling diagnosis of the *type* of representational collapse; (iii) it is lightweight, requiring only forward passes through two layers on 200 samples; and (iv) it is architecture-agnostic, applicable to any layer producing an activation matrix.

Future work will extend validation to larger datasets (ImageNet), modern architectures (ViTs, small LLMs), and adaptive compression frameworks where REPS components directly inform layer-wise compression decisions, enabling automated optimization of accuracy–efficiency trade-offs.

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
