# OpenReview forum: "DO CORESETS, PRUNING, AND QUANTIZATION PRESERVE NEURAL NETWORK REPRESENTATIONS?"
_ICLR.cc/2026/Workshop/GRaM — ICLR 2026 Workshop GRaM Poster_

### Official Review · Reviewer_vj6b · 2026-02-08
**Interesting empirical observations on compression, but lacks mathematical rigor and modern validation**

**Rating:** 5
**Confidence:** 4

**Review:**

The paper investigates the impact of coreset selection, pruning, and quantization on neural network representations. The authors propose two metrics, GTFA and REPS, to diagnose representational degradation. The work aims to provide a diagnostic tool that correlates better with accuracy drops than standard metrics.Strengths:Comprehensive Empirical Benchmark (Table 1): The breakdown of accuracy degradation across three different compression modalities (Coreset, Pruning, Quantization) in Table 1 is detailed and insightful. It clearly illustrates the "cliff" points where performance collapses (e.g., pruning > 50% or quantization < 4-bit). This provides a solid baseline for understanding how different compression techniques affect model stability.+1Motivation: The goal of developing a "compression-aware" diagnostic metric beyond simple test accuracy is well-motivated and important for the efficient deep learning community.Weaknesses:1. Mathematical Clarity and Rigor:While the empirical results are interesting, the mathematical formulation requires significant improvement to meet publication standards:Notation Ambiguity: In Eq. (11), the definition $s = \Sigma / ||\Sigma||_1$ is confusing. In the context of SVD, $\Sigma$ is a matrix, while $s$ is treated as a vector for entropy calculation. The authors should clarify whether they are referring to the vector of singular values.Formatting: There are minor typesetting errors (e.g., the stray "we" below Eq. (6)) that should be fixed.2. Justification of Metric Design:
The proposed REPS metric (Eq. 16) aggregates effective rank, aliveness, class separation, and eigenvalue decay. However, the specific weights ($\alpha=0.15, \beta=0.05$, etc.) appear to be heuristically tuned. The paper would benefit from a sensitivity analysis or a theoretical justification for why this specific linear combination is optimal/generalizable, rather than just fitted to the CIFAR-10 results.+13. Connection to "Geometry":The paper frames the contribution around "Geometry-grounded" learning. While the metrics use SVD (subspaces) and distances, the connection to the workshop's theme of topology or manifold geometry feels somewhat tenuous. The analysis is primarily statistical (variance, rank) rather than geometric in the topological sense. Softening these claims or strengthening the geometric interpretation would improve the paper.4. Experimental Scope:
The experiments are conducted on ResNet-18 and CIFAR-10. While Table 1 provides good insights for this setting, extending the validation to a more modern architecture (e.g., a ViT or a small LLM) or a harder dataset would significantly strengthen the claim that these metrics are robust tools for the current generation of models.Conclusion:The paper presents valuable empirical data on compression thresholds (Table 1) and proposes an interesting composite metric. However, the mathematical looseness and the reliance on heuristic weightings limit its current impact. I encourage the authors to refine the mathematical definitions and justify the metric coefficients to make this a stronger contribution.

**Pmlr Suitability:**

Yes

---

### Official Review · Reviewer_r6Ju · 2026-02-19
**Impact of compression methods in neural network representations**

**Rating:** 5
**Confidence:** 3

**Review:**

The idea of defining a metric that captures geometric perturbations induced compression is interesting and could improve the understanding of how such techniques affect generalization and robustness. However, the clarity and readability of the paper could be significantly improved. There are several typos (i.e lines 100-104 appear to be repeated), and some notation is not properly defined (like $\mathcal{M}$).

The choice of weights in equation (10) is not properly justified. In particular, is unclear to me why the weight of the weight similarity should be that much lower to functional similarity. This design choice seems important to the method and results, so I believe more theoretical insights or empirical evaluation are required. On the empirical side, I believe evaluating the method across a wider range of neural network architectures and datasets is necessary to justify the paper's claims.

Finally, the connection to the workshop topic is not clear. While symmetries of neural networks can be included into the framework through the similarity metrics, this direction is not really explored on the paper.

Overall, I believe the paper presents a nice idea, but it requires more theoretical end experimental exploration to be able to justify its claims.

**Pmlr Suitability:**

Yes

---

### Official Review · Reviewer_9Kwr · 2026-02-24
**Somewhat Interesting, But Implication Unclear**

**Rating:** 5
**Confidence:** 4

**Review:**

**Summary**: This paper studies how various model compression strategies: coresets, pruning, and quanitization affect model representations. They introduce two metrics: GTFA which measures how similar the weights/activation/etc are with the uncompressed model and Representation Similarity which studies more intrinsic properties of the representation space (i.e effective rank, seperability, etc).  They find that the Representation Similarity achieves better correlation to post-compression accuracy deteriorations and better than standard existing metrics.

**Strengths**: The paper appears to do careful analysis and examines a broad range of compression techniques.


**Weaknesses/Questions**:
-The provided metrics appear to be a combination of existing methods for studying the representational properties of neural networks and there is not much justification of why these metrics (as opposed to others) are used. Moreover, it is unclear to me why the weighting of these different metrics is performed as it is.
-  What is the practical usage of the metrics? The authors mention that they correlate well with degradation, but they don't justify whether their metrics are easier/less expensive than just measuring accuracy degradation.
- Only studied one model architecture: I think that the results would be much more convincing if they studied more than one architecture.
- Many of the results seem a bit unsurprising and the implications of them are not fully explained.

**Pmlr Suitability:**

No

---

### Meta-Review · Area_Chair_Zi3X · 2026-02-24

**Decision:**

Accept

**Metareview:**

The authors study the relationship between various neural compression methods and geometric perturbations in the representation space of neural networks. All the reviewers appreciated the work, and they found it interesting and relevant while having some concerns. I ask the authors to incorporate reviewers' comments carefully in the next version of their paper.

**Relevance To Proceedings:**

Yes — suitable for PMLR (long paper)

**Relevance To Workshop:**

Yes — suitable for GRaM

---

### Decision · Program_Chairs · 2026-03-02

Accept (Poster)